# Predictors of Symptom-Specific Treatment Response to Dietary Interventions in Irritable Bowel Syndrome

**DOI:** 10.3390/nu14020397

**Published:** 2022-01-17

**Authors:** Esther Colomier, Lukas Van Oudenhove, Jan Tack, Lena Böhn, Sean Bennet, Sanna Nybacka, Stine Störsrud, Lena Öhman, Hans Törnblom, Magnus Simrén

**Affiliations:** 1Department of Molecular and Clinical Medicine, Institute of Medicine, Sahlgrenska Academy, University of Gothenburg, 413 45 Gothenburg, Sweden; lena.bohn@vgregion.se (L.B.); sean.bennet@queensu.ca (S.B.); sanna.nybacka@gu.se (S.N.); stine.storsrud@vgregion.se (S.S.); hans.tornblom@gu.se (H.T.); magnus.simren@medicine.gu.se (M.S.); 2Translational Research Center for Gastrointestinal Disorders (TARGID), Department of Chronic Diseases and Metabolism (CHROMETA), KU Leuven, 3000 Leuven, Belgium; jan.tack@kuleuven.be; 3Laboratory for Brain-Gut Axis Studies (LaBGAS), Translational Research Center for GI Disorders, KU Leuven, 3000 Leuven, Belgium; lukas.vanoudenhove@kuleuven.be; 4Cognitive & Affective Neuroscience Lab (CANlab), Department of Psychological and Brain Sciences, Dartmouth College, Hanover, NH 03755, USA; 5Division of Gastroenterology, Department of Medicine, School of Medicine, Queen’s University, Kingston, ON K0G 1E0, Canada; 6Department of Microbiology and Immunology, Institute of Biomedicine, Sahlgrenska Academy, University of Gothenburg, 413 45 Gothenburg, Sweden; lena.ohman@microbio.gu.se; 7Center for Functional GI and Motility Disorders, University of North Carolina-Chapel Hill, Chapel Hill, NC 27599, USA

**Keywords:** prediction, treatment response, gastrointestinal symptoms, irritable bowel syndrome, disorders of the gut brain interaction, functional gastrointestinal disorders, diet, low FODMAP diet, NICE guidelines, dysbiosis

## Abstract

(1) Background: Predictors of dietary treatment response in irritable bowel syndrome (IBS) remain understudied. We aimed to investigate predictors of symptom improvement during the low FODMAP and the traditional IBS diet for four weeks. (2) Methods: Baseline measures included faecal Dysbiosis Index, food diaries with daily energy and FODMAP intake, non-gastrointestinal (GI) somatic symptoms, GI-specific anxiety, and psychological distress. Outcomes were bloating, constipation, diarrhea, and pain symptom scores treated as continuous variables in linear mixed models. (3) Results: We included 33 and 34 patients on the low FODMAP and traditional IBS diet, respectively. Less severe dysbiosis and higher energy intake predicted better pain response to both diets. Less severe dysbiosis also predicted better constipation response to both diets. More severe psychological distress predicted worse bloating response to both diets. For the different outcomes, several differential predictors were identified, indicating that baseline factors could predict better improvement in one treatment arm, but worse improvement in the other treatment arm. (4) Conclusions: Psychological, nutritional, and microbial factors predict symptom improvement when following the low FODMAP and traditional IBS diet. Findings may help individualize dietary treatment in IBS.

## 1. Introduction

Irritable bowel syndrome (IBS) is a disorder of gut-brain interaction (DGBI), formerly known as a functional gastrointestinal (GI) disorder, which affects 4% of the adult global population [1]. According to the diagnostic Rome IV criteria, IBS is characterized by weekly abdominal pain associated with altered bowel habits [2]. Apart from abdominal pain, altered stool frequency, diarrhea, and constipation, a large proportion of patients with IBS also experience other GI symptoms such as abdominal distention, bloating and flatulence [3]. The pathophysiology of IBS is only partly understood and multifactorial. Pathophysiological factors that are described up until today include altered gut-brain interactions, visceral hypersensitivity [4], abnormal GI motility [5,6], psychosocial factors, altered mucosal immune function and permeability [7,8,9], altered gut microbial environment with potential dysbiosis (imbalance in the microbiota homeostasis) [10,11], and food hypersensitivity [12].

The majority of patients with IBS report an association between food intake and symptoms [13,14]. A subset of patients with meal-related symptoms therefore severely self-restricts and avoids certain foods [15]. This distinct subgroup of patients may have a more severe symptom pattern, poorer quality of life, lower caloric intake, and reduced intake of several nutrients.

Due to the renewed focus on this meal-related factor in IBS, the interest in dietary treatment has increased. Many dietary interventions, such as a gluten-free, a low lactose, and a high fiber diet, as well as many more, have been studied in IBS [16]. However, the clinical guidelines currently focus primarily on two dietary treatment approaches. A first-line diet approach in IBS is to follow the traditional IBS diet described in the guidelines of the British Dietetic Association and National Institute for Health and Care Excellence (NICE) [17,18]. These guidelines include general lifestyle and dietary advice. They encourage physical activity, regular meal intake, a reduction of fat intake, and to avoid large meals, excessive insoluble fiber intake, and a few dietary triggers. Alternatively, or when patients do not respond to the traditional IBS diet or an initial pharmacological treatment approach, patients can be referred to a specialized dietician. They can then recommend a diet that limits the intake of poorly absorbed short-chain carbohydrates, i.e., fermentable oligo-, di-, monosaccharides, and polyols (FODMAPs)—the low FODMAP diet. In patients with IBS, symptoms can be triggered by the ingestion of these short-chain carbohydrates that are incompletely absorbed in the small intestine. FODMAPs are osmotically active, interfere with water absorption, and when entering the colon are precursors to gas production through fermentation by intestinal bacteria. This could cause luminal distention and altered motility, which in turn may lead to symptoms [19]. Randomized controlled trials show that approximately 50–75% of patients with IBS experience adequate symptom relief when following a low FODMAP diet [20,21,22]. Unfortunately, this leaves 25–50% of the patients with IBS not responding. Therefore, determining potential predictors of who would potentially benefit from dietary treatment and who would not, could be of major help to individualize the treatment approach.

Previous research has identified several potential predictors of response to the low FODMAP diet. A lower FODMAP intake before starting diet, a higher age, being female, and having fructose malabsorption was associated with responding better to the intervention [22,23]. Higher hydrogen levels after a nutrient challenge test were also predictive of a better response [24]. Specific volatile organic compound patterns found in the faeces of patients with IBS can separate responders from non-responders [25]. High baseline colonic methane and short-chain fatty acid production, high baseline symptom severity, and a high saccharolytic fermentation activity have also been identified as predictors of response [26,27]. Furthermore, since gut microbiota are involved in fermentation of FODMAPs, studies also focused on the predictive capacity of baseline faecal microbiota. Low FODMAP diet responders were characterized by a microbiota composition more closely related to the composition of a healthy reference group [28]. Specifically, responders presented a lower abundance of *Clostridia*, *Actinomycetales*, *Anaerotruncus*, and *Escherichia*, and a greater abundance of *Bacteroides fragilis *and* acidifiens, Acinetobacter, Ruminiclostridium, Streptococcus, Eubacterium, Streptococcus, Dorea,* and *Ruminococcus gnavus* [27,28,29,30]. One study even identified a gut microbial signature in IBS patients with diarrhea and mixed bowel habits that shows enhanced clinical responsiveness to the low FODMAP diet. The microbiomes in these patients were depleted in *Bacteroidetes* species and enriched in *Firmicutes* and genes for amino acid and carbohydrate metabolism [30]. However, there are also studies indicating that microbial factors do not predict response to the low FODMAP diet [21,31]. Studies that investigate the prediction of response to the traditional IBS diet are lacking. Apart from IBS-C patients and patients with a high saccharolytic fermentation activity being less likely to respond, no predictors of response to the traditional IBS diet have been described [23,27]. 

Most studies investigating predictors of treatment response dichotomized their outcome into responder vs. non-responder. However, treating the outcome as a continuous variable maximizes statistical power. Furthermore, response can also occur for one symptom, but not for the other. Therefore, our aim for this study was to investigate predictors of treatment response to the low FODMAP and the traditional IBS diet while focusing on the four core IBS symptoms, i.e., bloating, constipation, diarrhea, and pain as a post-hoc analysis of a previously published randomized controlled trial [23]. Hereby, treatment response was defined as the improvement of these symptoms during the intervention of four weeks. As potential predictors, we aimed to focus on different baseline factors that are of importance in the pathophysiology of IBS. We selected a psychological factor (i.e., psychological distress), non-GI somatic symptom severity, nutritional factors (i.e., energy, FODMAP, oligo-, di-, monosaccharide, and polyol intake), and microbial characteristics (i.e., an index that represents how severe the microbiota composition differs from the compositions of a healthy control sample). We studied psychological distress and non-GI somatic symptom severity, because the severity of both factors is closely related to more severe GI symptoms and may therefore influence the severity of symptom scores during the interventions [32]. Dietary variables were included because eating habits before the intervention might influence the dietary changes that the patients need to make during the intervention. Moreover, if patients are eating FODMAP restrictive prior to the intervention, their response could be dampened. Lastly, the microbial factor was added due to existing evidence that indicates that baseline feacal microbial composition together with fermentation activity influences the response to the dietary intervention [27,28,29].

## 2. Materials and Methods

### 2.1. Study Cohort

The study participants and general study methods that are not described here have been previously described in detail by Böhn et al. [23]. Briefly, we included patients fulfilling the Rome III criteria for IBS from the gastroenterology outpatient clinics in Gothenburg and Stockholm, Sweden. Patients with other GI diseases, such as coeliac disease or inflammatory bowel disease, severe liver, neurological, cardiac or psychiatric disease were excluded. Patients on excessively nutrient restrictive diets prior to the study were also not included. All patients provided written and verbal informed consent before the study start. 

### 2.2. Study Design

During a 10-day screening period, patients collected a faecal sample, completed a 4-day food diary, and a daily stool diary based on the Bristol Stool Form (BSF) scale [33] that was used to subgroup patients according to bowel habits. After these ten days, patients completed several GI and non-GI symptom questionnaires including the IBS severity scoring system (IBS-SSS) [34]. If the patients scored ≥ 175 corresponding to moderate or severe IBS, they were randomized to the low FODMAP diet or the traditional IBS diet by an online software. The allocated diet was then thoroughly explained by trained dieticians and followed for four weeks. During the intervention, patients weekly completed the symptom severity measure, the Gastrointestinal Symptom Rating Scale-IBS (GSRS-IBS) [35].

### 2.3. Dietary Interventions

#### 2.3.1. The Traditional IBS Diet

Patients randomized to the traditional IBS diet received instructions that were focused on ‘how’ and ‘when’ to eat. Patients needed to eat regularly, i.e., three meals and three snacks per day of adequate portion size. They needed to avoid feeling hungry or too full. Chewing thoroughly, eating in peace and quiet, and eating fibers (evenly distributed throughout the day) was encouraged. A few dietary products needed to be avoided or reduced (fatty and spicy foods, coffee, alcohol, onions, cabbage, beans, soft drinks and carbonated beverages, chewing gums, and sweeteners that ends with “-ol”).

#### 2.3.2. The Low FODMAP Diet

Patients randomized to the low FODMAP diet were instructed to limit the intake of all FODMAP containing foods [36]. Patients received a study specific brochure with detailed information about foods to avoid. The intake of foods rich in fructan (e.g., barley, rye, wheat, onion, leak, and garlic), galacto-oligosaccharides (GOS) (e.g., legumes and nuts), lactose (e.g., milk, yoghurt, and ice cream), fructose in excess of glucose (e.g., apples, honey, and orange juice), and polyols (e.g., avocado, celery, and artificial sweeteners) needed to be restricted. The brochure also contained lists of alternative foods that were allowed (e.g., lactose-free milk, gluten-free pasta, carrots, kiwi, and strawberries).

### 2.4. Dietary Intake

Food intake was recorded in 4-day food diaries in the week before the randomization and in the last week of the intervention period (Wednesday to Saturday). All food products and drinks were entered in the nutrient calculation software DIETIST XP version 3.1 (Kostdata.se, Stockholm, Sweden) added with an extensive FODMAP database [37]. For this analysis, we used the average daily intake of energy, FODMAP, oligosaccharide (i.e., GOS and fructan), disaccharide (i.e., lactose), monosaccharide (i.e., excess fructose), and polyols from the 4-day food diaries before the randomization. FODMAP intake was calculated by adding up the content of GOS, fructan, lactose, polyol, and excess fructose in every meal for four days. An average intake per day for all the dietary data was computed, which was then used to obtain the average daily intakes for all four days.

### 2.5. Questionnaires

#### 2.5.1. GI Symptoms

The primary outcome in this analysis was the improvement of four core IBS symptoms as measured by the GSRS-IBS, recorded retrospectively to reflect the last 7 days [35]. GSRS-IBS is a self-administered 13-items questionnaire with a 7-day recall period. Questions are scored on a 7-point Likert scale where one means ‘no symptoms’ and seven ‘severe symptoms’. There are 5 GSRS-IBS subscales with their respective sum score ranges: pain (2–14), bloating (3–21), constipation (2–14), diarrhea (4–28), and satiety (2–14). In order have an identical score range for all subscales, the average scores were considered for all subscales in this analysis (score ranges all 1–7). The satiety subscale, which is not considered as a key IBS symptom, was not included in this analysis. The other validated subscales thus assess the severity of four symptoms pivotal in the clinical picture of IBS. Due to our specific interest in symptom-based responses to the dietary interventions, we choose the GSRS-IBS subscales as our primary outcome instead of the validated response definition of >50 points reduction on the IBS-SSS, as information on individual key IBS symptoms cannot be assessed with the same level of detail with IBS-SSS.

#### 2.5.2. Non-GI Symptoms

Non-GI somatic symptoms were assessed with the Patient Health Questionnaire (PHQ)-12, which is a modified version of the PHQ-15 without three GI-related items [38]. The 12 items are scores on a scale from zero, ‘not bothered at all’, to two, ‘bothered a lot’. Summing up all symptom scores created a total score (women: 0–24, men: 0–22) with a high score representing a high non-GI somatic symptom burden during the last two weeks. One item assesses menstrual cramps/other period problems, leading to potentially higher PHQ-12 scores in women.

Psychological distress, i.e., anxiety and depression, was measured using the 14-item Hospital Anxiety and Depression scale (HADS) that has a one-week recall period [39]. Response scores range from zero to three that are used to create a sum score (0–42) with high scores representing more severe psychological distress.

GI-specific anxiety was assessed with the Visceral Sensitivity Index (VSI) [40], a 15-item questionnaire that contains statements related to fear of GI symptoms. Each item can be scored from one, ‘strongly agree’, to six, ‘strongly disagree’. High sum scores represent more severe GI-specific anxiety.

### 2.6. Gut Bacterial Analysis

Gut bacterial analysis was performed with the GA-map Dysbiosis Test (Genetic Analysis AS, Oslo, Norway) [41], which is a commercially available test that is described in more detail elsewhere [28]. In summary, the GA-map Dysbiosis Test created a bacterial profile and a Dysbiosis Index score ranging from one to five. A Dysbiosis Index score >2 indicates a microbiota composition that substantially differs from the composition of a healthy reference group and can thus be described as dysbiosis. In this analysis, we only used the Dysbiosis Index score as a potential predictor.

### 2.7. Data and Statistical Analysis

Statistical analysis was performed using IBM SPSS Statistics 26 software (SPSS Inc., Chicago, IL, USA) and SAS 9.4 software (SAS Institute, Cary, NC) for the univariate and multivariate analyses, respectively. Summary descriptive statistics are presented as mean ± standard deviation (SD) for continuous variables and *n* (%) for categorical variables. Significance was set at *p* < 0.05. Before performing the multivariate analysis, the distribution of all variables was tested for normality using the Shapiro–Wilk test. Non-normal distributions were transformed using Box Cox or log transformations [42]. The GSRS-IBS subscales used in the linear mixed (i.e., random-effect) models were BoxCox transformed. For the transformation of the GSRS-IBS subscales, the following BoxCox formula were used for all GSRS-IBS subscales: y(λ) = (y^λ^ − 1)/λ. The following specific formulas were used for pain, constipation, diarrhea and bloating subscale: y(1) = (y^1^−1)/1, y(0.75) = (y^0.75^ − 1)/0.75, y(0.25) = (y^0.25^ − 1)/0.25, and y(1) = (y^1^ − 1)/1, respectively. This led to changes in the original score range score of 1–7 to 0–6, 0–3.29, 0–2.51, and 0–6, respectively. Variables unable to achieve normal distributions were ordinalized into tertiles (i.e., Dysbiosis index score). 

To test our hypothesis that nutritional, microbial, and psychological factors could potentially predict symptom improvement when following a low FODMAP or traditional IBS diet, we used linear mixed (i.e., random-effect) models. Hereby, treatment response was not dichotomized, but GSRS-IBS subscale scores before, and during the intervention were used as continuous outcome variables. The models included the main effect of baseline predictors (testing the association between the predictor and GSRS-IBS score at the baseline timepoint) and the main effect of time (as a linear slope, one baseline timepoint and four during the intervention). Our main interest in this analysis was the included interaction effect between baseline predictor × time (testing the hypothesis that the baseline variable predicts the response slope). To test whether the baseline variables differentially predicted the response to the low FODMAP and traditional IBS diets, diet was added as a dichotomous variable to the models, including its main effect and all interactions, with the three-way baseline predictor × diet × time interaction effect testing this hypothesis.

## 3. Results

### 3.1. Clinical Characteristics

In total, 33 patients on low FODMAP diet and 34 patients on the traditional IBS diet were included in the analysis. Figure 1 provides further information about the included patients and which study measurements they completed. Demographic factors, distributions of IBS subtypes based on a BSF stool diary, baseline symptom severity, and baseline psychological, nutritional, and microbial factors of the two treatment arms are shown in Table 1.

### 3.2. Symptom Improvement during Dietary Interventions

In models with only the main effect of time as independent variable, both diets combined were shown to be effective at the group level for most symptoms. The dietary interventions significantly reduced the severity of pain, diarrhea, bloating (all *p* < 0.0001), but not of constipation (*p* = 0.15). Estimates can be found in Appendix A. When diet was added to the models, the results indicated absence of significant differences in response between both dietary interventions for any of the symptoms (pain: *p* = 0.80, constipation: *p* = 0.63, diarrhea: *p* = 0.96, bloating: *p* = 0.35). The models including diet and its interaction with time are displayed in Appendix A. This confirmed an earlier analysis on total IBS symptom severity assessed with the IBS-SSS by Böhn et al. [23].

### 3.3. Predictors of Pain Improvement

A higher energy intake significantly predicted better response to the low FODMAP and the traditional IBS diet when focusing on pain (time × energy intake interaction effect: *p* = 0.03), (Table 2). Figure 2a indicates that patients with a higher energy intake at baseline presented a significantly better pain improvement during the low FODMAP and the traditional IBS diet compared to patients with a lower energy intake. A lower Dysbiosis Index score (less severe dysbiosis) was associated with a better response to the diets (time × Dysbiosis Index (lowest vs. highest tertile) interaction effect: *p* = 0.05) (Table 2). Figure 2b shows the mean estimates over time of Dysbiosis Index scores grouped in tertiles. Patients with the lowest scores at baseline improved significantly more during the interventions compared to the tertile with the highest scores. When diet was included in the models, the intake of oligosaccharides was differentially associated with treatment response between both diets (time × oligosaccharide intake × diet interaction effect: *p* = 0.05) (Appendix A). This was driven by a significant association between higher baseline oligosaccharide intake and a better response to the traditional IBS diet, which was absent in the low FODMAP diet (time × oligosaccharide intake interaction effect in the low FODMAP diet and the traditional IBS diet: *p* = 0.49 and *p* = 0.02, respectively) (Figure 2c,d). All linear mixed models that contain the diet factor, including the models that did not identify significant differential predictors, are shown in Appendix A.

### 3.4. Predictors of Constipation and Diarrhea Improvement

When focusing on the improvement of constipation, a lower Dysbiosis Index score significantly predicted a better response to the dietary interventions (time × Dysbiosis Index (lowest vs. highest tertile) interaction effect: *p* = 0.01) (Table 2 and Figure 3a). When diet was added to the models, disaccharide (i.e., lactose) intake was identified as a differential predictor (time × disaccharide intake × diet interaction effect: *p* = 0.02), (Appendix A). This was again driven by a higher baseline disaccharide intake associated with a better response to the traditional diet, which was absent in the low FODMAP diet (time × disaccharide intake interaction effect in the low FODMAP diet and the traditional IBS diet: *p* = 0.46 and *p* = 0.01, respectively), (Figure 3b,c). For diarrhea, no predictors of treatment response were identified (Table 2). All linear mixed models that contain the diet factor, including the models that did not identify significant differential predictors, are shown in Appendix A.

### 3.5. Predictors of Bloating Improvement

More severe psychological distress significantly predicted worse response to the interventions (time × HADS interaction effect: *p* = 0.03) (Table 2 and Figure 4a). When diet was included in the models, oligosaccharide intake emerged as a differential predictor (time × oligosaccharide intake × diet interaction effect: *p* = 0.005), (Appendix A). Higher baseline oligosaccharide intake predicted worse response to the low FODMAP diet, but not to the traditional IBS diet (time × oligosaccharide intake interaction effect in the low FODMAP diet and the traditional IBS diet: *p* = 0.01 and *p* = 0.16, respectively), (Figure 4b,c). All linear mixed models that contain the diet factor, including the models that did not identify significant differential predictors, are shown in Appendix A.

## 4. Discussion

This study evaluated potential predictors of symptom response to the traditional IBS and the low FODMAP diet in patients with IBS. Instead of grouping participants in responders and non-responders, we used the improvement of four core IBS symptoms during the intervention weeks as our outcome. We showed that a microbial and dietary factor predicted better pain improvement due to the interventions. The same microbial factor predicted better constipation improvement when following both dietary regimes. We also identified a psychological factor that was associated with worse bloating improvement during the dietary interventions. Multiple dietary factors were also found to be differential predictors, meaning that they predicted either better or worse symptom improvement during one treatment, but not to the other.

When studies investigate treatment response predictors, a straightforward predictor that can be identified is the level of baseline symptom severity. A previous study found that patients who respond to the low FODMAP diet indicate higher baseline symptom severity scores when completing the IBS-SSS questionnaire [27]. However, in our sample, there were indications that patients with less severe clinical characteristics responded better to the treatments. Patients with a higher energy intake at baseline and a lower Dysbiosis Index score, i.e., a score more closely related to that of healthy controls, improved more than patients with a lower energy intake and a higher Dysbiosis Index score at baseline. A recent study showed that patients with a more severe symptom pattern, but still a normal BMI, have a lower energy intake possibly due to food avoidance and restriction [15]. Our findings suggest that patients who have a lower energy intake and who have a microbial composition that substantially differs from that of healthy controls might need different treatment approaches. When Vervier et al. described a gut microbial signature that predicted a better response to a low FODMAP intervention, they indicated the opposite [30]. Microbiota profiles that resembled a more pathogenic phenotype had a greater reduction in IBS-SSS after the low FODMAP diet compared to patients with a profile more closely related to healthy controls. However, it is not clear whether the differences between our findings could be attributable to methodological differences. Our patient population (i.e., specific IBS subtypes vs. all subtypes), but also our primary outcome, and the way microbial factors were studied differed. Based on our results, the patients with a more severe disease might not have adequate symptom relief when they follow the traditional IBS or low FODMAP diet alone. They may potentially benefit from more elaborate treatment approaches potentially using different combinations of diet, peripherally and centrally acting pharmaceutical interventions and behavioral therapies. 

Furthermore, patients with more severe psychological distress appear to respond worse to both diets. It has been shown that patients who demonstrate more severe psychological distress also report more severe GI symptoms in general [32]. Patients with more severe psychological distress, that could also partly be food-related, might be hypersensitive and potentially more skeptical towards modifying their diets. However, a survey including 256 patients with IBS indicated that most IBS patients (around 80%) prefer receiving pharmacological as well as dietary therapy [43]. Another possible explanation for the worse response in patients with more severe psychological distress could therefore again be that these patients might need combinations of pharmacological, behavioral, and other treatment options. Physicians should consider the potential food-related psychological distress that these patients might experience. When Peters et al. compared the effect of gut-directed hypnotherapy, the low FODMAP diet or a combination therapy, no significant differences in efficacy were found between the treatments [44]. However, in this study, there were no differences in the psychological measurements at baseline between the treatment arms. What they did report is that hypnotherapy resulted in superior improvements of the psychological measures. When patients thus report severe psychological symptoms, it can be hypothesized that the food-related psychological distress may need to be targeted first before making dietary changes in order to alleviate symptoms.

Our study also indicated that it is possible to identify baseline characteristics that are of importance for the response to one diet, but not for the other diet. These differential predictors included the intake of different FODMAP groups at baseline. Our findings might indicate that when it comes to dietary advice, we need to stick to the originally proposed treatment algorithm by recommending the traditional IBS diet first and only proceeding with the low FODMAP diet when this does not adequately relieve symptoms. In this study, patients who have higher baseline intakes of different FODMAP groups responded well to the traditional IBS diet. This observation was not present in the treatment arm that followed the low FODMAP diet. In fact, patients with higher oligosaccharide intake at baseline had a worse bloating response to the low FODMAP diet. Patients who have not identified FODMAPs as triggers and do not avoid and restrict them in daily life might not benefit from a diet as restrictive as the low FODMAP diet. For these patients following the traditional diet and applying lifestyle changes proposed by the British Dietetic Association and the NICE guidelines could already adequately relieve symptoms. Even if the low FODMAP diet shows efficacy in IBS, there are also negative consequences of this diet, such as deterioration of the Dysbiosis Index score, and reduction of potentially beneficial bacteria, such as *Bifidobacterium* [28,45,46]. Therefore, it is recommended to start with the less restrictive first-line therapy, i.e., the traditional IBS diet, and if this does not adequately relieve symptoms, only then proceed with the second-line treatment, i.e., the low FODMAP diet.

The key strength of our study is the novel approach to not categorize the patients in two groups, i.e., responders and non-responders, but to focus on individual core IBS symptoms. In order to develop suitable treatment algorithms, we do believe that considering the heterogeneity of IBS symptom patterns is of importance. One patient might be considered a non-responder based on an inadequate reduction of the IBS-SSS total score, but the patient could have experienced substantial improvement of one specific bothersome IBS symptom. Furthermore, we preselected potential predictors from three different domains that are highly important in the pathophysiology of IBS, i.e., psychological, microbial and nutritional factors. Nevertheless, this study also has limitations including the use of a convenience sample. When designing the initial study, the calculation of the sample size was not based on this analysis. Additionally, we conducted this study with a limited number of patients appointed to two treatment groups. Moreover, preselecting potential predictors of importance was based on the limited number of studies that have already looked into predictors of treatment response together with the study data that were available.

Since the current treatment options in IBS remain symptom-based with efficacy levels that are often limited, developing predictive evidence-based treatment algorithms remains a priority for future research. Repeating this analysis in larger samples, powered for this approach, should be encouraged. Future studies should consider including potential predictors that could be easily measured in clinical practice and assess more treatment options than the ones we addressed. Investigators should preferably select a practical and effective treatment response measure that considers the heterogeneity of IBS symptom patterns.

In conclusion, this study shows that patterns of psychological, dietary, and microbial factors can predict IBS symptom response to two dietary advice systems—the traditional IBS and the low FODMAP diet. A clinical profile with less severe IBS features appears to predict a better symptom-specific response to both dietary interventions. These findings should be confirmed in larger prospective randomized controlled trials, preferably including different treatment approaches that have been shown to be efficacious in IBS. Results may help to optimize personalized treatment algorithms in IBS.

## Figures and Tables

**Figure 1 nutrients-14-00397-f001:**
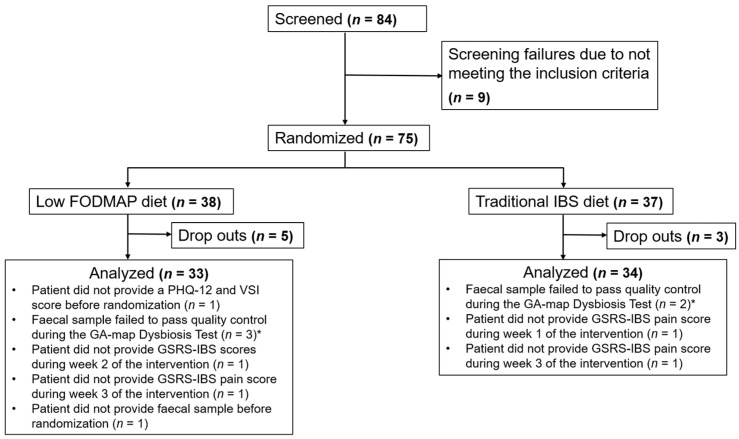
Flow chart indicating the number of patients included in each study phase, for details see text. Fermentable oligo-, di-, monosaccharides, and polyols, FODMAP; Patient Health Questionnaire-12, PHQ-12; Visceral Sensitivity Index, VSI; Gastrointestinal Symptom Rating Scale-Irritable Bowel Syndrome, GSRS-IBS.

**Figure 2 nutrients-14-00397-f002:**
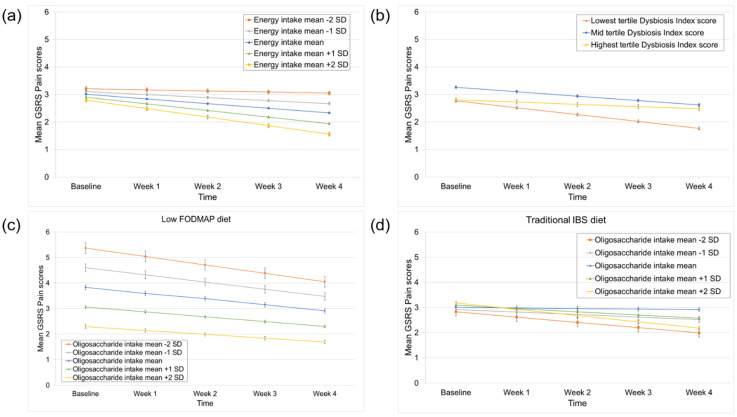
Predictors of pain improvement during the low FODMAP and traditional IBS diet. (**a**) Higher energy intake predicted better response to both diets. (**b**) Lower Dysbiosis Index scores predicted better response to both diets. (**c**) Oligosaccharide intake was a differential predictor with a non-significant association between higher baseline oligosaccharide intake and a worse response in low FODMAP treatment arm and (**d**) a significant association between higher baseline oligosaccharide intake and a better response in traditional IBS treatment arm. The GSRS-IBS subscales used in the linear mixed (i.e., random-effect) models were BoxCox transformed. For the transformation of the pain subscale, the following BoxCox formula was used: $y\left(\lambda \right)=\frac{y^{\lambda }-1}{\lambda }$. This led to changes in the original score range score of 1–7 to 0–6, using the formula: $y\left(1\right)=\frac{y^{1}-1}{1}$.

**Figure 3 nutrients-14-00397-f003:**
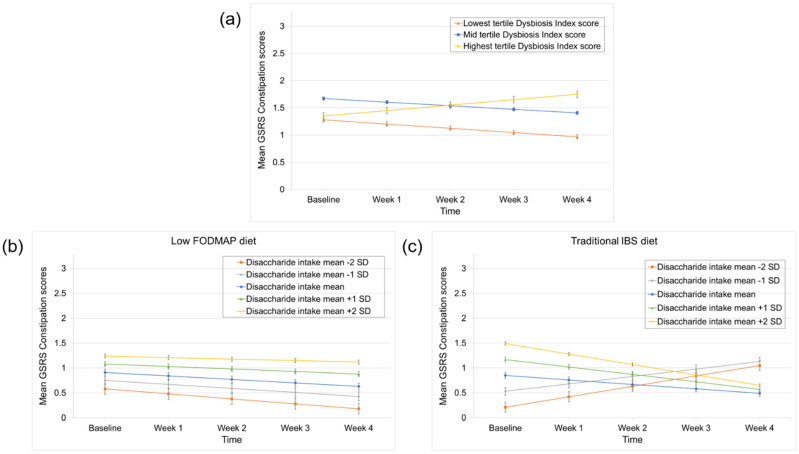
Predictors of constipation improvement during the low FODMAP and traditional IBS diet. (**a**) A lower Dysbiosis Index score predicted better response to both diets. (**b**) Disaccharide intake was a differential predictor with a non-significant association between higher baseline disaccharide intake and a worse response in low FODMAP treatment arm and (**c**) a significant association between higher baseline disaccharide intake and a better response in traditional IBS treatment arm. The GSRS-IBS subscales used in the linear mixed (i.e., random-effect) models were BoxCox transformed. For the transformation of the constipation subscale, the following BoxCox formula was used: $y\left(\lambda \right)=\frac{y^{\lambda }-1}{\lambda }$. This led to changes in the original score range score of 1–7 to 0–3.29, using the formula: $y\left(0.5\right)=\frac{y^{0.5}-1}{0.5}$.

**Figure 4 nutrients-14-00397-f004:**
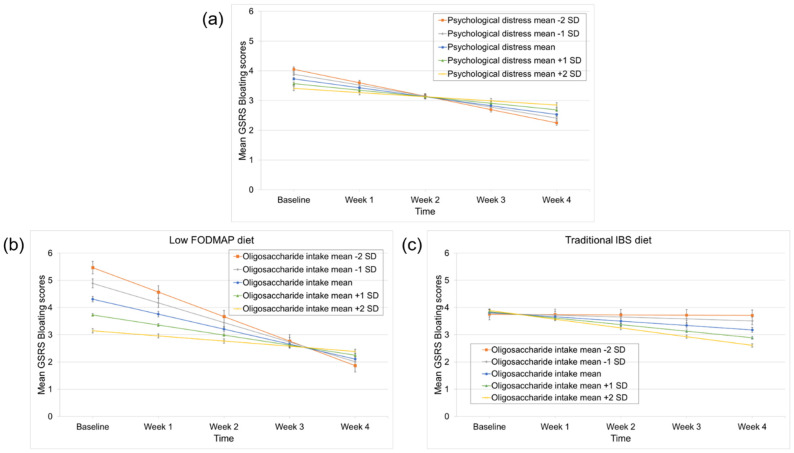
Predictors of bloating improvement during the low FODMAP and traditional IBS diet. (**a**) Higher psychological distress predicted worse response to both diets. (**b**) Oligosaccharide intake was a differential predictor with a significant association between higher baseline oligosaccharide intake and a worse response in low FODMAP treatment arm and (**c**) a non-significant association between higher baseline oligosaccharide intake and a better response in traditional IBS treatment arm. The GSRS-IBS subscales used in the linear mixed (i.e., random-effect) models were BoxCox transformed. For the transformation of the bloating subscale, the following BoxCox formula was used: $y\left(\lambda \right)=\frac{y^{\lambda }-1}{\lambda }$. This led to changes in the original score range score of 1–7 to 0–6, using the formula: $y\left(1\right)=\frac{y^{1}-1}{1}$.

**Table 1 nutrients-14-00397-t001:** Clinical characteristics of the two treatment arms at baseline *.

Clinical Characteristic	Dietary Intervention	
	Low FODMAP Diet	Traditional IBS Diet
Gender (%) (*n* = 67)		
Female	85.8	83.6
Male	15.2	16.4
Age (years) (*n* = 67)	45 ± 15	40 ± 17
BMI (kg/m^2^) (*n* = 65)	24.39 ± 3.93	24.07 ± 3.75
IBS subtypes (%) (*n* = 67)		
IBS-C	27.3	31.3
IBS-D	30.3	28.4
IBS-nonCnonD	42.4	40.3
IBS-SSS (score) (*n* = 67)	317 ± 70	306 ± 66
GSRS-IBS total (score) (*n* = 67)	3.91 ±0.95	3.64 ± 0.75
GSRS-IBS pain	4.05 ± 1.00	4.13 ± 0.93
GSRS-IBS bloating	4.98 ± 1.08	5.13 ± 1.38
GSRS-IBS constipation	3.03 ± 1.57	3.29 ± 1.65
GSRS-IBS diarrhea	3.41 ± 1.01	3.29 ± 1.12
PHQ-12 (score) (*n* = 66)	8.16 ± 3.89	7.47 ± 3.40
VSI (score) (*n* = 66)	49.13 ± 12.83	47.59 ±16.85
HADS total (score) (*n* = 67)	13.55 ± 7.39	10.74 ± 6.74
FODMAP intake (gram/day) (*n* = 67)	18.49 ± 9.90	18.49 ± 7.92
Oligosaccharide intake (gram/day)	2.72 ± 1.21	2.87 ± 1.42
Disaccharide intake (gram/day)	8.80 ± 8.33	8.47 ± 5.69
Monosaccharide intake (gram/day)	5.91 ± 5.42	6.03 ± 5.49
Polyols intake (gram/day)	1.07 ±1.56	1.12 ± 1.44
Energy intake (Kcal/day) (*n* = 67)	2097 ± 430	2100 ± 451
Dysbiosis index score (% 1/2/3/4/5) (*n* = 60)	10.3/20.7/48.3/10.3/10.3	6.7/16.7/53.3/13.3/10.0

* NOTE: Irritable bowel syndrome (IBS) with predominant constipation, IBS-C; IBS with predominant diarrhea, IBS-D; IBS-non-constipation, non-diarrhea, IBS-nonCnonD; IBS severity scoring system, IBS-SSS; gastrointestinal symptom rating scale for IBS, GSRS-IBS; patient health questionnaire-12, PHQ-12; visceral sensitivity index, VSI; hospital anxiety and depression scale, HADS; fermentable oligo-, di-, monosaccharides, and polyols, FODMAP.

**Table 2 nutrients-14-00397-t002:** Mixed linear random-effect models without diet effects and GSRS-IBS subscales as outcome *.

	Pain			Constipation			Diarrhea			Bloating		
Independent variables: PHQ-12 and time	β	SE	*p*-value	β	SE	*p*-value	β	SE	*p*-value	β	SE	*p*-value
(intercept)	3.02	0.11	**<0.0001**	1.48	0.11	**<0.0001**	1.32	0.05	**<0.0001**	3.72	0.15	**<0.0001**
Time	−0.18	0.03	**<0.0001**	−0.03	0.02	0.18	−0.06	0.01	**<0.0001**	−0.30	0.04	**<0.0001**
PHQ-12	0.37	0.11	**0.001**	0.01	0.11	0.91	0.13	0.05	**0.01**	0.24	0.15	0.11
Time × PHQ-12	−0.01	0.03	0.67	0.01	0.02	0.60	−0.008	0.01	0.54	0.01	0.04	0.70
Independent variables: HADS and time	β	SE	*p*-value	β	SE	*p*-value	β	SE	*p*-value	β	SE	*p*-value
(intercept)	3.01	0.12	**<0.0001**	1.48	0.11	**<0.0001**	1.33	0.05	**<0.0001**	3.73	0.15	**<0.0001**
Time	−0.17	0.03	**<0.0001**	−0.03	0.02	0.15	−0.06	0.01	**<0.0001**	−0.30	0.04	**<0.0001**
HADS	0.03	0.12	0.80	−0.10	0.11	0.37	0.03	0.05	0.55	−0.16	0.15	0.28
Time × HADS	0.01	0.03	0.70	0.01	0.02	0.75	0.004	0.01	0.72	0.08	0.04	**0.03**
Independent variables: VSI and time	β	SE	*p*-value	β	SE	*p*-value	β	SE	*p*-value	β	SE	*p*-value
(intercept)	3.02	0.11	**<0.0001**	1.48	0.11	**<0.0001**	1.32	0.05	**<0.0001**	3.72	0.15	**<0.0001**
Time	−0.18	0.03	**<0.0001**	−0.03	0.02	0.18	−0.06	0.01	**<0.0001**	−0.30	0.04	**<0.0001**
VSI	0.29	0.11	**0.01**	0.17	0.11	0.11	0.14	0.05	**0.01**	0.11	0.15	0.47
Time × VSI	0.01	0.03	0.68	−0.001	0.02	0.97	−0.01	0.01	0.60	0.06	0.04	0.11
Independent variables: FODMAP intake and time	β	SE	*p*-value	β	SE	*p*-value	β	SE	*p*-value	β	SE	*p*-value
(intercept)	3.01	0.12	**<0.0001**	1.48	0.10	**<0.0001**	1.33	0.05	**<0.0001**	3.73	0.15	**<0.0001**
Time	−0.17	0.03	**<0.0001**	−0.03	0.02	0.15	−0.06	0.01	**<0.0001**	−0.30	0.04	**<0.0001**
FODMAP intake	0.02	0.12	0.83	0.18	0.10	0.09	0.05	0.05	0.33	−0.12	0.15	0.41
Time × FODMAP intake	0.01	0.03	0.80	0.01	0.02	0.71	−0.01	0.01	0.69	0.001	0.04	0.97
Independent variables: Energy intake and time	β	SE	*p*-value	β	SE	*p*-value	β	SE	*p*-value	β	SE	*p*-value
(intercept)	3.01	0.12	**<0.0001**	1.48	0.11	**<0.0001**	1.33	0.05	**<0.0001**	3.73	0.14	**<0.0001**
Time	−0.17	0.03	**<0.0001**	−0.03	0.02	0.15	−0.06	0.01	**<0.0001**	−0.30	0.04	**<0.0001**
Energy intake	−0.10	0.12	0.39	0.01	0.11	0.94	−0.06	0.05	0.24	−0.34	0.14	**0.03**
Time × Energy intake	−0.07	0.03	**0.03**	−0.01	0.02	0.72	−0.02	0.01	0.20	0.01	0.04	0.81
Independent variables: Dysbiosis Index (DI) score (in tertiles) and time	β	SE	*p*-value	β	SE	*p*-value	β	SE	*p*-value	β	SE	*p*-value
(intercept)	2.80	0.25	**<0.0001**	1.35	0.23	**<0.0001**	1.42	0.11	**<0.0001**	3.72	0.32	**<0.0001**
Time	−0.08	0.07	0.24	0.01	0.05	**0.05**	−0.07	0.03	**0.01**	−0.25	0.08	**0.003**
DI score (lowest vs. highest tertile)	−0.03	0.32	0.92	−0.08	0.30	0.79	−0.25	0.15	0.09	−0.34	0.41	0.41
DI score (mid vs. highest tertile)	0.45	0.30	0.13	0.31	0.28	0.26	−0.02	0.13	0.87	0.25	0.39	0.51
DI score (lowest vs. mid tertile)	−0.49	0.26	0.06	−0.39	0.24	0.11	−0.22	0.12	0.06	−0.59	0.34	0.08
Time × DI score (lowest vs. highest tertile)	−0.17	0.09	**0.05**	−0.18	0.06	**0.01**	0.03	0.04	0.36	−0.06	0.10	0.59
Time × DI score (mid vs. highest tertile)	−0.08	0.08	0.31	−0.17	0.06	**0.01**	0.01	0.03	0.81	−0.05	0.10	0.62
Time × DI score (lowest vs. mid tertile)	−0.09	0.07	0.20	−0.01	0.05	0.81	0.03	0.03	0.40	−0.008	0.09	0.92
Independent variables: Oligosaccharide intake and time	β	SE	*p*-value	β	SE	*p*-value	β	SE	*p*-value	β	SE	*p*-value
(intercept)	3.34	0.22	**<0.0001**	1.50	0.20	**<0.0001**	1.29	0.10	**<0.0001**	4.00	0.28	**<0.0001**
Time	−0.12	0.06	0.06	−0.04	0.05	0.33	−0.04	0.02	0.11	−0.33	0.07	**<0.0001**
Oligosaccharide intake	−0.27	0.15	0.07	−0.02	0.14	0.91	0.04	0.07	0.60	−0.22	0.19	0.25
Time × Oligosaccharide intake	−0.05	0.04	0.25	−0.01	0.03	0.80	−0.01	0.02	0.42	0.03	0.05	0.53
Independent variables: Polyols intake and time	β	SE	*p*-value	β	SE	*p*-value	β	SE	*p*-value	β	SE	*p*-value
(intercept)	3.07	0.14	**<0.0001**	1.50	0.13	**<0.0001**	1.39	0.06	**<0.0001**	3.67	0.19	**<0.0001**
Time	−0.17	0.04	**<0.0001**	−0.04	0.03	0.15	−0.06	0.02	**0.001**	−0.26	0.05	**<0.0001**
Polyols intake	0.05	0.08	0.47	0.02	0.07	0.79	0.05	0.04	0.16	−0.06	0.11	0.56
Time × Polyols intake	0.0001	0.02	>0.99	−0.01	0.02	0.61	0.001	0.01	0.93	0.03	0.02	0.21
Independent variables: Disaccharide intake and time	β	SE	*p*-value	β	SE	*p*-value	β	SE	*p*-value	β	SE	*p*-value
(intercept)	2.62	0.22	**<0.0001**	0.89	0.19	**<0.0001**	1.27	0.10	**<0.0001**	3.70	0.30	**<0.0001**
Time	−0.13	0.06	**0.04**	0.01	0.05	0.82	−0.05	0.03	0.07	−0.30	0.07	**0.0003**
Disaccharide intake	0.16	0.08	**0.05**	0.24	0.07	**0.001**	0.03	0.04	0.48	0.01	0.10	0.92
Time × Disaccharide intake	−0.02	0.02	0.45	−0.02	0.02	0.26	−0.004	0.001	0.68	−0.01	0.03	0.80
Independent variables: Monosaccharide intake and time	**β**	SE	*p*-value	β	SE	*p*-value	β	SE	*p*-value	β	SE	*p*-value
(intercept)	3.05	0.19	**<0.0001**	1.59	0.18	**<0.0001**	1.32	0.09	**<0.0001**	3.81	0.25	**<0.0001**
Time	−0.18	0.05	**0.001**	−0.07	0.04	0.08	−0.04	0.02	**0.03**	−0.25	0.06	**0.0001**
Monosaccharide intake	−0.04	0.11	0.75	−0.07	0.10	0.49	0.003	0.05	0.96	−0.06	0.14	0.67
Time × Monosaccharide intake	0.004	0.03	0.89	0.03	0.02	0.22	−0.01	0.01	0.64	−0.03	0.04	0.35

* NOTE: Gastrointestinal symptom rating scale for IBS, GSRS-IBS; patient health questionnaire-12, PHQ-12; visceral sensitivity index, VSI; hospital anxiety and depression scale, HADS; fermentable oligo-, di-, monosaccharides, and polyols, FODMAP; dysbiosis index, DI; effect size, β; standard error, SE; the GSRS-IBS subscales used in the linear mixed (i.e., random-effect) models were BoxCox transformed. The bold indicates the significant *p*-values in this large table.

## Data Availability

The data presented in this study are available on request from the corresponding author. The data are not publicly available due to privacy restrictions.

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
