# Peer review of "Predictors of Symptom-Specific Treatment Response to Dietary Interventions in Irritable Bowel Syndrome"

_nutrients, 2022, doi:10.3390/nu14020397_

Round 1
Reviewer 1 Report
This is a well-written paper on the predictors of clinical response to two different dietary modifications (low-FODMAP vs. traditional IBS diet). The results are of interest to many clinicians and IBS researchers. Literature is well covered and up-to-date. However, the paper contains huge amount of data and tables/figures, making the paper too detailed and difficult to read. However, this challenge is difficult to circumvent. The discussion is versatile  and insightful, but some improvement would be beneficial there too.Â
First to address:
- Reduce the amount of data in the text, and transfer some of the less important findings into the supplements.Â
- IBS-SSS was originally chosen as primary outcome of the study (Böhn 2015). However, the authors chose to select another assessment, i.e. GSRS-IBS. Why was this done? The question raises, if the results on IBS-SSS did not become significant. It is important to present the predictor data using IBS-SSS as well.
- The paper by Vervier et al. (Gut 2021, doi: 10.1136/gutjnl-2021-325177) should be drawn into light latest in the discussion, as their results seem to contradict at least partly the present results of the authors regarding dysbiosis.Â
- The discussion is strong on diet but weaker on behavioral therapy/psychology. Please, reflect how effective behavioral interventions are vs. dietary in IBS, and do we have indications that psychological interventions help more those that are more affected by psychosocial burden.Â
- Did the lower BMI predict worse response? You suggest that patients who eat less, smaller portions, do not respond as well (lower energy intake). In clinic, you often get an impression that severely affected patients have low BMI.Â
- Extend the discussion on the limitations of the study, post-hoc nature, etc.Â
- Underline 2-3 major findings once more in the conclusion.Â
Secondary comments.Â
- Why you did not select physical exercise habits as a potential predictor? it is well known that regular physical exercise help in keeping symptoms in better control; patients with strong PE habits may respond  worse, or better.Â
- The Monash study on hypnotherapy vs. low-FODMAP would be valuable addition to the discussion. Peters et al. 2016
- Did you stratify data according to severity of IBS at baseline using IBS-SSS? Would patients with IBS-SSS over 300 respond poorly, as they seemed to when using GSRS? As far as I know, clinical severity is primarily assessed by IBS-SSS.Â
- Did you monitor the use of peppermint oil, probiotics, glutamate, insoluble fiber products, loperamide, and use of other OTC products?Â
- Did you analyze meal rhythm; i.e. 3 mains and 3 snacks/day vs. irregular  dietary pattern?
Â
Reviewer 2 Report
The manuscript is well written and I read with interest your manuscript. The only limitation of your study is the small number of patients included (33 and 34 patients in the low FODMAP and traditionally IBS diet).
